# Evaluation of Genetic Diversity in Dog Breeds Using Pedigree and Molecular Analysis: A Review

Ripfumelo Success Mabunda [1,2], Mahlako Linah Makgahlela [2,3], Khathutshelo Agree Nephawe [1,*] and Bohani Mtileni [1,*]

1 Department of Animal Sciences, Tshwane University of Technology, Pretoria 0001, South Africa
2 Agricultural Research Council, Animal Production, Irene 0062, South Africa
3 Department of Animal, Wildlife and Grassland Sciences, University of the Free State, Bloemfontein 9301, South Africa
* Correspondence: nephaweka@tut.ac.za (K.A.N.); mtilenib@tut.ac.za (B.M.)

**Abstract:** Domestic dogs are important for many economic and social reasons, and they have become a well-known model species for human disease. According to research, dog breeds exhibit significant levels of inbreeding and genetic diversity loss, decreasing the population's ability to adapt in certain conditions, and indicating the need of conservation strategies. Before the development of molecular markers, pedigree information was used for genetic diversity management. In recent years, genomic tools are frequently applied for accurate estimation of genetic diversity and improved genetic conservation due to incomplete pedigrees and pedigree errors. The most frequently used molecular markers include PCR-based microsatellite markers (STRs) and DNA sequencing-based single-nucleotide polymorphism markers (SNP). The aim of this review was to highlight genetic diversity studies on dog breeds conducted using pedigree and molecular markers, as well as the importance of genetic diversity conservation in increasing the adaptability and survival of dog breed populations.

**Keywords:** genetic conservation; genetic diversity; microsatellite markers; pedigree information; single-nucleotide polymorphism markers

## 1. Introduction

Dogs are descended from the grey wolf (*Canis lupus*) and were domesticated in Southeast Asia around 33,000 years ago [1,2]. Canine ancestors migrated alongside humans to Africa and the Middle East around 15,000 years ago, and then to Europe around 10,000 years ago [3].

Domestic dog breeds (*Canis familiaris* L.) were the earliest domesticated animals, with over 400 breeds around the world [4], and more than 150 being bred in Korea [5,6]. These dog breeds have evolved into a broad collection of breeds with diverse morphological and physiological features through domestication, and natural and artificial selection [1]. Some are stray dogs throughout the world, mostly in villages and cities near humans [7]. Domestication of a canine ancestor is very likely to have occurred during first settlements and early agriculture and is accepted as a first population bottleneck during which the size of genetic variation within the population was effectively reduced [8,9]. Regardless of the precise timeframe, it is undeniable that for many years, people have been selecting for certain behavioural traits such as peacefulness, agreeableness, non-aggressiveness, and loyalty, as well as physical characteristics such as coat colour, coat length, height, and facial appearance [10,11].

Dogs have evolved into one of the most common domestic species, as well as the most common carnivore. Their global population is estimated to be close to 900 million, and it is undoubtedly growing [12,13]. Modern dog breeds differ from other domesticated species due to the large amount of phenotypic variation caused by human selective desire [14],

and because of their intelligence and behavioural abilities. The dog has evolved into hundreds of races with various variations, ranging from the Chihuahua's height of a few tens of centimetres to the Irish Wolfhound's height of more than one meter [15]. Thus, their appearance and behaviour are significantly different [16].

The development of a new generation of domestic dog breeds entirely dependent on genetic variation as observed through genetic differences within and between breeds [17]. However, the process of domestication has resulted in genetic bottlenecks, which have impacted the evolution of modern dog breeds [4]. Genetic bottlenecks are evolutionary events that cause a random decrease in the genetic variation, resulting in small founding populations and genetic drift [18]. The first bottleneck occurred >15,000 years ago during the domestication from the grey wolf [19,20]. The second bottleneck was caused by the isolation of the current dog breeds during the past >300 years, resulting in a smaller number of potential parents. The third bottleneck was from the use of popular sires after an intense selection for exterior traits [21].

Furthermore, most dog breeds are closed populations with no gene flow from outside, and only a small number of dogs are utilized for breeding [22,23], resulting in a loss of genetic diversity within and between breeds [24], and inbreeding with the occurrence of genetic defects and depression in fitness traits. Therefore, genetic variation management is necessary to prevent high levels of inbreeding, the loss of genetic diversity, and the emergence of genetic disorders in small populations [25]. The aim of this review was to highlight genetic diversity studies on dog breeds conducted using pedigree and molecular markers, as well as the importance of genetic diversity conservation in increasing the adaptability and survival of dog breed populations.

## 2. The Importance of Keeping Dogs

The dog has been man's faithful companion throughout history. They help with daily activities and make their families happy [26]. Dogs can be a source of comfort in times of emotional difficulty, as well as having positive psychological and physical health impacts [27]. Different behavioural responses, or temperaments, have resulted from the diversity of dog breeds, making dogs ideal for a variety of roles ranging from pets to working dogs. However, the owner's expectations of the dog may differ from what the dog is required to provide [28]. It is, therefore, the owner's responsibility to give appropriate care to the dog, and to realize that it is a descendant of the wolf and should have the possibility to show its natural behaviour.

Some dogs are calm in nature, while others are aggressive, which changes their utility as they perform many roles for mankind [29]. Law enforcement uses service and working dogs to assist the police and military, while government agencies use them for a variety of purposes such as explosives and drug detection [30]. They were used for a wide range of purposes, including heading, pulling loads, therapy, sports activities, medical and genomics research, customs, rescue, security work, identifying biological material, companionship [6,31–34], as a fighter, hunter, hauler, and source of food and fur [13]. Dogs are also used to help people with disabilities, such as guide dogs for the blind, seizure alert dogs, and hearing dogs [35,36]. People have also used dogs in specialty positions in which their superior sense of smell has been used to seek out termites, missing persons, and, in some instances, malignant tumours, due to their ability to learn and be directed by humans [11].

Ownership of a dog is likely to benefit the owner's physical and mental health, including decreased depression, increased oxytocin levels, and lower blood pressure and cholesterol levels [37,38]. Dogs also encourage their owners to exercise on a regular basis, lowering the risk of cardiovascular disease for both parties [39]. Exercising with their owner is likely the primary source of exercise for a companion animal, and therefore it is strongly influenced by owner-related factors such as their physical and social environment, personal capabilities, interests, motivating factors for exercise, and relationship with their dog [40]. Dog owners promote social contact among themselves, which reduces feelings

of loneliness [41]. Owning a dog has been shown to have positive psychological impacts, including less stress effects, a larger social communication network, and a high sympathetic ability and sense of mercy. Furthermore, people who live with dogs are less likely to become ill because of their gratifying life with the dog [42]. Companion animals are also ideal research subjects for investigating the genetic and environmental factors that influence human behaviour, personality, and psychiatric diseases [43].

Domestic dogs are important for many economic reasons [44]. Pet dogs are the driving force behind a multibillion sector that includes food production, veterinary care, specialty services, and, of course, dog breeding [45]. Breeding and selling dogs accounts for 5% of all dog-related income in Belgium and is a major source of employment [46]. However, because of genetic improvement, the focus of breeding shifted away from working ability and toward morphological characters (such as coat colour, texture, size, and skull shape) [28]. Consequently, this phenotypic selection of around 400 breeds is now regarded as the second bottleneck [28]. Therefore, conserving dogs with special abilities in these situations will preserve the genetics underlying them, allowing for their continued use and study [47].

The United Kingdom Kennel Club recognizes 215 dog breeds and separates them into seven divisions by function. Hound group hunting dogs are those that hunt by scent and sight. Gundog track (Pointers and Setters), hunt (Spaniels), and recover game (Retrievers). The terrier category consists of canines that were used to hunt vermin or foxes. Utility dogs were originally employed for working or guarding but are now largely companions. Working dogs were used for house guarding, and hunting. Pastorals herd and guard livestock [48]. Because of small body sizes, toy dogs are mostly kept as pets [48].

## 3. The Importance of Conserving Genetic Diversity in Dog Breeds

Effective management of domestic animal resources requires a full knowledge of breed characteristics, such as population size and structure, geographical distribution, genetic diversity within and between breeds [49], and the combination of different alleles into haplotypes that characterizes different breeds. According to Ocampo et al. [50], genetic diversity refers to the presence of diverse alleles or genes in a population, which is represented in physical, physiological, and behavioural differences between individuals and populations. Genetic diversity is essential for animal species' survival and improvement [51]. The goal of population structure assessment is to investigate the occurrence of possible changes in genetic variability distribution within and between subpopulations that make up the population, as well as the rate at which genetic diversity is lost. Populations that have been subjected to selective pressure over time tend to change in their original structure [52]. To assess population changes over a short period of time, parameters based on the probability of genetic origin from different herds [53], founders [54], and ancestors [55] were used [56]. A population's genetic structure is shaped by various mechanisms such as gene flow, selection pressure, mutation, and genetic drift, which varies with time and is influenced by several internal and external factors to the population [57].

Dogs are an excellent model for studying genetic variation and population recovery, both of which are major concerns in conservation genetics [58]. Since the early 1800s, when modern breeding practices became popular, genetic diversity in many dog breeds has been continuously decreasing [59]. With continued biodiversity loss and an increase in the number of threatened and extinct species [60], science is becoming increasingly important in nature and wildlife conservation. As a result, initiatives to save nature and species are of global and fundamental importance to humanity [61].

Many old domestic breeds that are not being used in large-scale commercial production have small populations, and some are considered endangered [32]. According to Kettunen et al. [62], the Norwegian Lundehund is a critically endangered indigenous dog breed, with the current population having lost 38.8% of its genetic diversity. When the Second World War broke out, the population of Norwegian Lundehund had reduced in size to just fifty individuals [62]. The breed's inbreeding depression is indicated by low fertility and a high frequency of genetic susceptibility to intestinal disorder. Therefore,

the only way to ensure the survival of this endangered breed is to introduce individuals from other breeds as breeding candidates [62]. Furthermore, to preserve genetic diversity, Norwegian Lundehund is outcrossed with Norwegian Buhund, Icelandic Sheepdog, and Norrbottenspets. These three breeds were chosen based on phenotypes and then genetically analysed to increase Lundehund genetic diversity and reduce health problems [63].

The Swedish Board of Agriculture has also found several traditional Swedish breeds of conservation concern, including ten dog breeds, namely Hamilton hound, Norrbottenspitz, Schiller hound, Smaland hound, Swedish elkhound, Swedish lapphound, Danish-Swedish farmdog, Drever, Gotland hound and Swedish vallhund in terms of domestic animal populations [59]. Furthermore, rare, and national breeds, such as Polish Hounds [64] or Ca Mè [65], are commonly the most affected by genetic diversity loss. The Czech Spotted Dog (CSD) is no exception [66]. According to Navas et al. [66], the Ca Mè dog breed has long been at risk of genetic diversity loss due to a long process of crossing with foreign breeds that has contributed to the loss of its genetic identity since the 1950s. However, this is a common framework for many breeds, as many are characterized by a reduced genetic diversity due to a small number of founders in their process of development [66]. Therefore, introducing dogs of other breeds or unknown origin that are phenotypically similar to a given breed is one way to enrich a limited gene pool [64].

The preservation of genetic diversity is a primary goal of population management in breeding and conservation programs [67]. Species diversity provides resistance, strength, stability, and liveliness to the system [68]. It is critical to assess a breed's genetic variability for future use [69], and it is useful for introducing appropriate selection and conservation methods in dog breeds [1,2]. Domestic animal populations have received increased conservation genetic attention over the years [70]. This focus includes both scientific research and national and international policy initiatives [70].

Furthermore, if genetic variability is maintained, special traits of domestic dog breeds that are not known today may be important in the future [68]. According to Lee et al. [5], it is critical to value indigenous genetic resources for breed conservation. FAO [71] reported that selection should preserve breeds as genetically and culturally unique genetic resources. Selection for increased production while ignoring traits associated to conservation traits such as adaptation, specific genetic variants, and product quality can reduce breed uniqueness and between-breed variation [72]. Therefore, there is a need to put effective management strategies in place to ensure preservation of genetic diversity in dog breeds.

## 4. The Management of Genetic Diversity

Genetic characterisation within breeds is essential for assessing a dog breed's future reproductive potential [73]. The genetic structure of the population assists in monitoring gene flow by giving information on the founding ancestors of the population and their contributions to current population's genetic variability [74]. Animals with high levels of inbreeding (>6.25%) can be prevented from breeding, while animals with kinships below a specific level can only be mated [25], to maintain populations' genetic diversity.

Current population management guidelines for endangered breeds prioritise genetic uniqueness over diversity [75]. The Committee on Biological Diversity's Strategic Plan target 13 [76] aims to "prevent genetic loss" and "protect genetic diversity" of socioeconomically and culturally significant species [31,58]. Using pedigree information, the influence of a breeding management strategy on genetic diversity changes may be assessed [77,78]. Molecular markers have also recently been developed to manage genetic resources by accurately estimating genetic uniqueness and controlling inbreeding [79,80].

### 4.1. Pedigree Information

When kept properly, studbooks provide complete information on a population and are useful for analysing genetic diversity and population structure [81]. In comparison to using molecular data, a pedigree is the simplest and most cost-effective way to assess genetic diversity and demographic parameters of a population in recent years [82–84]. Aside from

controlling inbreeding, understanding the population structure is essential for effectively implementing and monitoring a selection program [85]. As a result, a significant loss of genetic diversity in populations can be avoided [86].

Using pedigree information to characterize genetic diversity allows researchers to determine the population's structure and changes through time [69], as well as the breed's history and breeding techniques [87]. Pedigree data could be included into genetic programming (GP) models to account for residual polygenic variance not captured by markers and to minimize empirical bias in predictions [88]. Pedigree data can also be used to calculate the probability of gene origin (the genetic contribution of founders and ancestors), as well as to determine effective population size, and historical bottlenecks in the population [50,77,89].

The completeness of pedigrees has a significant impact on the quality of inbreeding estimation as well as the evaluation of inbreeding depression [67]. Accurate estimates of population parameters are important [73], with the likelihood of finding common ancestors increasing with the level of completeness of the pedigree [90]. Puente et al. [91] found that the pedigree completeness (PCI) for ca de Rater and ca de Bestiar dog breeds was lower at 60% for the fifth generation than that reported by Leroy et al. [92], having a PCI of 100% for the fifth generation for internationally recognized breeds. These results may be due to their endangered condition and the absence of genetic management in both breeds, as formal structures were only recently recognized, or that both breeds' genetic management programs are still in the early phases of development [91]. Navas et al. [65], reported higher inbreeding values for Ca Mè dog breed, which were backed up by significantly higher levels of PCI over generations, which could be due to the larger importance of pedigree information. According to Velie et al. [93], since 1933 the Working Kelpie Council of Australia has maintained the breed's pedigrees with open-registry and allowed dogs of unknown parentage to be registered for breeding purposes. As a result, the Australian working kelpie population is ideally suited for exploring inbreeding levels in an open-registry pedigreed dog breed [93]. The ability to record individuals of unknown origin in pedigree books may aid in increasing the genetic diversity of such breeds [81].

Although pedigree data is useful for assessing genetic diversity, it is sometimes limited by missing or incorrect data [52,94]. As a result, the analysis is limited to a few generations, which corresponds to the beginning of computerized pedigree recording or within the breed [87]. Pedigree analyses also fail to estimate correct genotypic probabilities, which are often too complex to compute even on modern computers [32]. Inbreeding levels measured by pedigree analysis are highly dependent on the accuracy and quality of pedigree data [95]. As a result, incomplete pedigree information may affect the average coefficient of inbreeding, and close relationships between some individuals may not be true [90,96]. However, even when unknown sires are missing from the pedigree, Van Raden's method for measuring inbreeding coefficients, combined with statistics based on the probability of gene origin, are adequate for assessing genetic diversity in populations [97].

According to Ceh and Dovc [57], the effective population sizes in the Karst Shepherd were 19.3 and 98.3 in the Tornjak and concluded that the latter results were overestimated due to incomplete pedigree information. Furthermore, Mortlock et al. [94] found that the Bullmastiff reference population's pedigree data had a higher inbreeding coefficient of 0.047 than the molecular data (0.035). However, the molecular estimate of inbreeding was similar to the entire pedigree database estimate (0.039), implying that molecular approaches can detect values representative of the overall population in the absence of complete pedigree information and the importance of keeping pedigree information up to date. Furthermore, the presence of pedigree errors causes bias, which has been reported to be 1 to 10% [23], and lack of knowledge on ancestry or important ancestors not detected may bias the results; thus, pedigree errors must be avoided at all costs [98,99]. As a result, studies on dog breeds must be conducted for both pedigree and genomic studies, to determine how well pedigree results correspond to genomic inbreeding and levels of genetic variation [59], to ensure the successful conservation of the dog breeds.

Individuals' genetic relatedness can be determined using both pedigree information and genome-wide markers [88]. Genomic data give observed and realized relationships, whereas pedigrees represent expected average relationships defining potential gene transfer [88]. Using both pedigree and genomic information, on the other hand, may yield superior results in terms of genetic similarity between genotypes [88]. Furthermore, because pedigree analysis typically picks up recent events that influence diversity, such as recent inbreeding and use of popular breeding individuals [94], the level of genetic diversity measured using molecular methods may differ from that measured by pedigree analysis. Molecular markers can detect recent inbreeding (runs of homozygosity) and trace it back to the potential effects of selection, migration, and previous breeding events, which can be traced back further than pedigree analysis, which are limited by the start of pedigree recording and the quality of records [94].

Table 1 presents the genetic diversity of dog breeds as determined by pedigree information. The equivalent complete generations (EqG) ranged from 2.14 to 10.88, with Finnish Spitz in Finland having the highest EqG (10.88) [100] and Bichon Frise in Belgium having the lowest EqG (2.14) [23]. High levels of pedigree completeness in Finnish Spitz dogs indicate that effort was taken in pedigree recording, as pedigree completeness is important for estimating accurate inbreeding levels, whereas low levels of EqG underestimate the levels of inbreeding in populations. As a result, efforts in animal recording must be made to ensure the reliability of pedigree information.

Inbreeding levels ranged from 0.039 to 44.7, with Bouvier des Ardennes having the highest inbreeding coefficient (*F*) (44.7) in a Belgian study [23] and Bullmastiff having the lowest *F* (0.039) in a study conducted in Australia by Mortlock et al. [94]. This suggests that mating occurred between related animals for Bouvier des Ardennes. However, given that Bouvier des Ardennes has a small effective population size (*Ne*) (3.2), these findings are to be expected.

The number of ancestors contributing > 50% of genetic diversity to the population ranged from 2 to 31. Polish Lowland Sheepdogs for the study conducted in Germany [101] and Czech Spotted dog in Czech Republic [67] had only two ancestors contributing over 50% of the populations' genetic variability, compared to the high number of ancestors who contributed less, implying a loss of genetic diversity. More ancestors contributing to the population's genetic variability was found in the Ca de Rater population in Spain [91], indicating increased genetic diversity.

The effective number founders ($f_e$) to effective number of ancestors ($f_a$) ratio ranged from 1 to 7.3. According to Boichard et al. [55], the $f_e$ to $f_a$ ratio must be one for founders to contribute the same genetic diversity in the population, with a ratio higher than one indicating a genetic bottleneck or reduced population size. The Border Collie dog [89] in research from Australia was the most affected by genetic bottleneck ($f_e / f_a = 7.3$) because of past incidents and through selection, compared to the Bouvier des Ardennes, which had not yet been impacted ($f_e / f_a = 1$) due to its recent re-establishment [23].

The effective population size (*Ne*) ranged from 3.2 to 384.62. According to FAO [71], the *Ne* should be between 50 to 100. The Ca de Bestiar [91] research in Spain had the highest level of *Ne* (384.62), showing that there are still effective breeding animals to preserve the population's genetic variability. The Bouvier des Ardennes had the lowest *Ne* (3.2) [23], indicating a loss of effective breeding animals and genetic diversity, limiting the breed's adaptability and survival.

**Table 1.** Genetic diversity of dog breeds assessed using pedigree information.

| Breed | Country | EqG | F | N > 50% | $f_e$ | $f_a$ | $f_e/f_a$ | Ne | References |
|-------|---------|-----|---|---------|-------|-------|-----------|-----|------------|
| Bracco Italiano | Italy | 4.70 | 4.10 | 9 | 61.3 | 46 | 1.3 | 38.86 | [102] |
| Tatra Shepherd | Poland | 3.44 | 7.17 | 4 | 44 | 11 | 4 | - | [103] |
| Bullmastiff | Australia | 3.24 | 0.039 | 20 | 79 | 62 | 1.3 | 41 | [94] |
| Bichon frise | Belgium | 2.14 | 10.0 | - | 13 | 10 | 1.3 | 17.8 | [23] |
| Bouvier des Ardennes | Belgium | 4.87 | 44.7 | - | 3 | 3 | 1 | 3.2 | [23] |
| Finnish Spitz | Finland | 10.88 | 6.33 | - | 30.71 | 20.18 | 1.5 | 73.53 | [101] |
| Nordic Spitz | Finland | 6.54 | 4.36 | - | 42.35 | 27.91 | 1.5 | 108.70 | [101] |
| Border Collie | Hungary | 4.47 | 9.86 | 8 | 117 | 20 | 5.85 | >100 | [24] |
| Polish Hunting | Poland | - | 0.1151 | - | - | - | - | 28.51 | [104] |
| Ca de Bestiar | Spain | - | 0.34 | 15 | 87.32 | 26 | 3.4 | 384.62 | [91] |
| Ca de Rater | Spain | - | 1.41 | 31 | 66.08 | 36 | 1.8 | 54.35 | [91] |
| Ca Mè | Spain | 4.87 | 11.23 | - | 29.09 | 10 | 2.9 | 13.25 | [65] |
| Polish Lowland Sheepdogs | Germany | 10.09 | 0.18 | 2 | 10 | 6 | 1.7 | 22.16 | [101] |
| Czech Spotted | Czech Republic | 9.46 | 36.45 | 2 | 4 | 3 | 1.3 | 10.28. | [67] |
| Border Collie | Australia | 10.4 | 0.095 | 13 | 205.5 | 28 | 7.3 | 123.5 | [89] |
| Deutsch Drahthaar | Germany | 8.62 | 0.042 | 13 | 65.5 | 37.8 | 1.7 | 91.6 | [105] |

EqG—Equivalent complete generations; F—Average inbreeding coefficient; N > 50%—Number of ancestors contributing > 50% genetic variability; $f_e$—Effective number of founders; $f_a$—Effective number of ancestors; *Ne*—Effective population size.

Table 2 presents some of the tools available for measuring genetic diversity in pedigree analysis. PEDIG software includes several independent algorithms that determine gene origin, relationship, and inbreeding coefficients, as well as characterise pedigree information [106]. It is well-suited to large-scale population study (up to several tens of millions of individuals). The PEDIG programme was used to examine genetic diversity in the Bullmastiff dog breed [94], as well as purebred Irish Wolfhounds [98].

According to Gutiérrez and Goyache [107], the programme ENDOG uses pedigree information to measure individuals' inbreeding and average relationship coefficients, effective population size, and parameters characterizing probability of gene origins, such as the effective number of founders and ancestors. ENDOG was used to examine 75,389 records. The Ca Mè dog breed [65], Ca de Rater, and Ca de Bestiar's genetic diversity was measured using ENDOG software [91].

The Coancestry, Inbreeding (*F*), and Contribution (CFC) programme is used to measure individuals' inbreeding coefficients and relationships, founder genome equivalent, and effective number of non-founders in pedigree analysis [108]. Average coancestry and individual inbreeding coefficients have been measured for 1,010,500 individuals [108]. The CFC was used to estimate *F* in the Braque Français type Pyrénées [109] and the Basset Hound dog population [110].

PyPedal is a pedigree analysis software that offers a variety of genetic diversity measures, including inbreeding and relationship coefficients, effective number of founders and ancestors, and founder genome equivalents [111]. Inbreeding estimates have been evaluated for over 500,000 individuals. PyPedal was verified using dairy cow and working dog pedigrees [111].

Population Management x (PMx) software includes project notes, population characterization (Selection), demography, genetics, evaluation of population goals, and documenting recommendations for which animals to breed. The programme requires no more than 20,000 individuals [112]. The PMx tool was used to assess genetic diversity in traditional dog breeds [59] and Polish Greyhounds [81].

**Table 2.** Programs used to measure genetic diversity in pedigree analysis.

| Programs | Uses | Data Records | Download | References |
|---|---|---|---|---|
| PEDIG | Pedigree completeness, inbreeding and relationship coefficients, effective founders, and ancestors. | Several tens of million individuals. | https://www6.jouy.inrae.fr/gabi_eng/Support-Expertise/Software/Pedig (accessed on 26 October 2022) | [106] |
| ENDOG | Inbreeding and relationship coefficients, effective founders and ancestors, and generation intervals. | 75,000 records. | http://www.ucm.es/info/prodanim/html/JP_Web.htm#_Endog_3.0:_A. (accessed on 26 October 2022) | [107] |
| Coancestry, inbreeding (*F*) and Contribution (CFC) | Inbreeding coefficients and relationships, founder genome equivalent, and effective non-founders. | 1,010,500 individuals. | https://mybiosoftware.com/cfc-1-0-monitor-genetic-diversity.html (accessed on 26 October 2022) | [108] |
| PyPedal | Pedigree completeness, inbreeding and relationship coefficients, effective founders, and ancestors. | ~500,000 animals for inbreeding calculations. | https://pypedal.sourceforge.net/ (accessed on 26 October 2022) | [111] |
| Population Management x (PMx) | Inbreeding coefficients, kinship, and founder allele contribution and survival. | 20,000 individuals. | https://scti.tools/downloads/#tab-4200585ee2aed8893e8 (accessed on 26 October 2022) | [112] |

### 4.2. Molecular Markers

The molecular markers are Deoxyribonucleic Acid (DNA) sequences in a particular region of the genome that are inherited according to the Mendelian principle, encode or do not always encode certain features, and are unaffected by the environment [113]. These DNA-based markers are used to identify genetic variation within and between populations as well as to map numerous genes across many species [114–117].

The DNA marker information has dominated categorization of species in recent years and has established itself as a trustworthy tool in population genetics, phylogenetic relationships, metagenomics, and ancestry studies [118,119]. According to Kumar et al. [120], a good marker should detect many alleles, be repeatable, error-free, provide the correct information at each run, and be inexpensive. The most frequently used molecular markers nowadays are microsatellite markers (STRs) based on Polymerase Chain Reaction (PCR) and single-nucleotide polymorphism markers (SNP) based on DNA sequencing [121–123].

#### 4.2.1. Microsatellite Markers

Microsatellite Markers (STRs) are nucleotide tandem repeats that typically comprise 1 to 10 base pair (bp) unit motifs [124,125]. Because of their great polymorphism, STRs are commonly used in population genetics and genetic ancestry [126]. They have long been recognised as valuable tools for evaluating genetic diversity and population change in dog breed populations [17,34,81], as well as parentage verification [34,127,128]. STRs may be found throughout the genome of eukaryotes and have a high mutation rate.

Table 3 summarizes the molecular genetic diversity of dog breeds using microsatellite markers. The expected heterozygosity ($H_E$) ranged from 0.38 to 0.77, whereas the observed heterozygosity ($H_O$) ranged from 0.37 to 0.816. According to a study in Italy [49], Maremma Sheepdogs had the highest $H_E$ (0.77) when using 18-STR markers. Using 21-STR markers, the Polish Greyhound in Poland was found to have the lowest $H_E$ (0.38) [81]. The highest $H_O$ (0.816) was found in African painted dogs for research conducted in the United States of America [129] using 14-STR markers. The Polish Greyhound had a lower $H_O$ (0.37) [81]. These findings suggest that the Polish Greyhound dog population has reduced genetic diversity, as seen by low levels of $H_E$ and $H_O$ when compared to other breeds.

**Table 3.** Molecular genetic diversity on dog breeds using microsatellite markers.

| Breed | Country | $H_E$ | $H_O$ | $F_{IS}$ | STR Panel | References |
|---|---|---|---|---|---|---|
| Jack Russell terrier | United Kingdom | 0.76 | 0.75 | 0.016 | 15-STR | [130] |
| Gyeongju Donggyeong | Republic of Korea | 0.7266 | 0.7657 | - | 10-STR | [5] |
| German Shepherd | Pakistan | - | 0.7420 | −0.5864 | 15-STR | [127] |
| Labrador retriever | Pakistan | - | 0.6754 | −0.50 | 15-STR | [127] |
| Oropa Shepherd | Italy | 0.62 | 0.70 | −0.14 | 18-STR | [49] |
| Maremma Sheepdog | Italy | 0.77 | 0.69 | 0.11 | 18-STR | [49] |
| Bouvier des Ardennes | Belgium | 0.668 | 0.714 | −0.040 | 19-STR | [23] |
| Rottweiler | Belgium | 0.534 | 0.536 | −0.004 | 19-STR | [23] |
| English bulldogs | United States of America | 0.575 | 0.573 | 0.007 | 33-STR | [131] |
| Rough-haired Segugio Italiano | Italy | 0.722 | 0.680 | 0.056 | 21-STR | [132] |
| Short-haired Segugio Italiano | Italy | 0.716 | 0.689 | 0.036 | 21-STR | [132] |
| Tatra Shepherd | Poland | 0.643 | 0.645 | −0.0046 | 18-STR | [133] |
| Polish Hunting | Poland | 0.6050 | 0.6142 | −0.012 | 21-STR | [104] |
| Cesky Fousek | Czech Republic | 0.673 | 0.669 | 0.005 | 18-STR | [134] |
| Yugoslavian Shepherd | Serbia | 0.728 | 0.696 | 0.041 | 9-STR | [135] |
| African painted dogs | United States of America | 0.746 | 0.816 | −0.108 | 14-STR | [129] |
| Polish Greyhound | Poland | 0.38 | 0.37 | −0.018 | 21-STR | [81] |
| Yorkshire Terrier | Poland | 0.698 | 0.662 | 0.0533 | 21-STR | [34] |
| Irish Wolfhound | Poland | 0.474 | 0.491 | −0.0373 | 21-STR | [34] |
| Ca Rater Mallorquí | Spain | 0.685 | 0.655 | 0.044 | 33-STR | [136] |

$H_E$—Expected heterozygosity; $H_O$—Observed heterozygosity; $F_{IS}$—Inbreeding coefficients from marker genotypes.

The inbreeding coefficients from marker genotypes ($F_{IS}$) ranged from $-0.004$ to 0.11, with the Maremma Sheepdog in Italy having the highest $F_{IS}$ (0.11) for 18-STR markers [49]. In research conducted in Belgium [23], the Rottweiler had a negative $F_{IS}$ ($-0.004$) for 19-STR markers. The negative $F_{IS}$ in the Rottweiler breed indicates low levels of inbreeding and avoidance of mating related animals, whereas the positive $F_{IS}$ in the Maremma Sheepdog indicates high levels of inbreeding and mating between related animals.

### 4.2.2. Single-Nucleotide Polymorphism

A single nucleotide polymorphism (SNP) is a DNA locus or place at which different individuals within a species differ [137]. For instance, in most individuals, the G nucleotide occurs at a certain base location in the genome, whereas an A appears in a tiny percentage of the population. This implies that a SNP exists at this position, and the two possible nucleotide changes (G or A) are known as the alleles for this location [138]. SNPs are abundant in the genome, have a lower mutation rate than STR markers and Mitochondrial DNA (mtDNA), and can be measured and analysed consistently once discovered [139]. They are the most prevalent sequence variations in the genome and were just recently discovered to be helpful tools for measuring genetic diversity [122,123].

The SNP data are widely used for determining marker-trait associations in genomewide association studies (GWAS), creating high-resolution genetic maps [140,141], assessing population evolutionary history through landscape genomics [142], and genomic selection [143–145]. Using measures of heterozygosity, SNP markers have also been utilised in dog breeds to assess genetic diversity [89,146]. Additionally, runs of homozygosity (ROH) using SNP data can be used to estimate inbreeding in diploid individuals, especially when pedigree information is lacking, inaccurate, or unavailable [147].

Table 4 presents the molecular genetic diversity of dog breeds using single nucleotide polymorphism markers. The expected ($H_E$) and observed heterozygosity ($H_O$) ranged from 0.035 to 0.38 and 0.038 to 0.4, respectively. In research conducted China, the Mongolia Xi had the highest $H_E$ (0.38) when using the 170K SNP chip [148]. Using a 170K SNP chip, the Norwegian Lundehund in Norway had the lowest $H_E$ (0.035) [149]. The highest $H_O$ (0.4) was observed in research performed in the Republic of Korea for Korean Jindo White [150] using a 173,662K SNP chip and Mongolia Xi ($H_O = 0.4$) [148]. The lowest $H_O$ (0.038) was found in the Norwegian Lundehund [149]. Low levels of $H_E$ and $H_O$ were found in Norwegian Lundehunds when compared to other breeds, indicating a loss of genetic variability that may impact the breed's adaptability and survival.

**Table 4.** Molecular genetic diversity on dog breeds using single nucleotide polymorphism markers.

| Breed | Country | $H_E$ | $H_O$ | $F_{ROH}$ | $F_{IS}$ | SNP Panel | References |
|---|---|---|---|---|---|---|---|
| Norwegian Lundehund | Norway | 0.035 | 0.038 | - | - | 170K SNP chip | [149] |
| Korean Jindo White | Republic of Korea | 0.32 | 0.4 | - | −0.22 | 173,662K SNP chip | [150] |
| Braque Français, type Pyrénées | Italy | 0.359 | 0.371 | 0.112 | −0.127, 0.172 | 170K SNP chip | [1] |
| Mongolia Xi | China | 0.38 | 0.4 | 0.08 | - | 170K SNP chip | [148] |
| Shanxi Xi | China | 0.30 | 0.31 | 0.28 | - | 170K SNP chip | [148] |
| Sapsaree | Republic of Korea | - | 0.342 | - | - | 20K SNP chip | [2] |
| Old English Sheepdog | Republic of Korea | - | 0.179 | - | - | 170K SNP chip | [2] |
| Istrian shorthaired hound | Republic of Croatia | 0.311 | 0.317 | 0.123 | −0.006 | 220K SNP chip | [151] |
| Bracco Italiano | Republic of Croatia | 0.253 | 0.268 | 0.248 | −0.001 | 220K SNP chip | [151] |
| Border Collie | Australia | 0.328 | 0.309 | 0.037 | - | 170K and 220K SNP chips | [89] |

$H_E$—Expected heterozygosity; $H_O$—Observed heterozygosity; $F_{ROH}$—Genomic inbreeding estimated from runs of homozygosity; $F_{IS}$—Inbreeding coefficients from marker genotypes.

Inbreeding assessed from runs of homozygosity ($F_{ROH}$) ranged from 0.037 to 0.28, with Shanxi Xi having the highest $F_{ROH}$ (0.28) in China [148] using a 170K SNP chip. The Border Collie in Australia had the lowest $F_{ROH}$ (0.037) using 170K and 220K SNP chips [89].

Inbreeding coefficients assessed from marker genotypes ($F_{IS}$) ranged from −0.001 to 0.172, with the Braque Français, type Pyrénées having the highest $F_{IS}$ (0.172) in Italy [1] using a 170K SNP chip. A negative $F_{IS}$ (−0.001) value was found for Bracco Italiano in the Republic of Croatia using a 220K SNP chip [151]. The low $F_{ROH}$ value in Border Collie dogs and negative $F_{IS}$ value in Bracco Italiano dogs indicate effective management strategies that prevent mating of related animals.

## 5. Factors Contributing to the Loss of Genetic Diversity

The loss of genetic diversity in small populations is a serious concern for conservation around the world because it can diminish fitness and adaptability, which can lead to breed extinction [58]. Breeders' mating strategies such as inbreeding and the use of popular sires have a direct impact on the structure of the breed and the loss of founder alleles [73]. Small population sizes and mating decisions simply based on desired phenotypes (appearance and behaviour) without regard for relatedness are characteristic features of dog breeding [152]. Many factors contribute to the loss of genetic diversity and adaptability, and to understand this loss, particularly in small populations, it is necessary to understand processes such as genetic bottleneck and random genetic drift that have occurred in the population [24,68].

### 5.1. The Founders' Genetic Contribution

Animals with unknown parents are referred to as founders, and they directly contribute to the gene pool of the reference population [153]. The effective number of founders ($f_e$) are the number of founders which contribute equally and produce the same genetic diversity in the populations under consideration [50]. The effective number of ancestors ($f_a$) are the small number of ancestors (founders or not) required to explain the population's entire genetic diversity [154]. At the start of the management programme, it is critical to have as many individuals as possible, since these include all the genetic variability that can be conserved [155]. Wijnrocx et al. [23] revealed a low number of founders (3) in a re-established Bouvier des Ardennes dog population, which has resulted in higher *F* (44.7), indicating genetic diversity loss.

Genetic variability can be reduced because of unequal contributions from founders and genetic bottlenecks [156]. According to Machová et al. [67], the bottleneck effect occurs when a small number of animals are chosen for breeding or when a population is subdivided. When a small number of sires may provide a large amount of genetic variety compared to the rest of the sires, genetic diversity is lost. According to Oliveira et al. [67], only a few animals contributed 50% of within-breed genetic variability in their study, which could become a concern if the sire usage policy remains unchanged. Kania-Gierdziewicz et al. [103] reported four ancestors to have contributed 50% of the population's genetic variability compared to 40% contributed by twenty ancestors in Tatra Shepherd dog population. Additionally, Ács et al. [24] reported that only eight founders were able to contribute 50% of the population's genetic variability in the Border Collie dog breed, suggesting that there is an unequal contribution of genetic variability by founders.

Higher or lower values of the $f_e/f_a$ ratio from one usually suggest an uneven usage of sires, which puts the original genetic diversity at danger [67]. Therefore, when an unequal number of males and females in a population is severe, there will be a reduction in the number of animals available for reproduction, which is one of the main factors responsible for a loss of genetic variability [52]. Loss of genetic diversity was reported for the Border Collie in Hungary, with a high $f_e/f_a$ ratio of 5.85 [24]. Furthermore, in Australia, Soh et al. [89] reported an extremely high $f_e/f_a$ ratio of 7.3 in Border Collie dogs, implying a severe reduction in genetic diversity due to small population size.

*5.2. Reduction in Effective Population Size*

According to Broeckx [99], the effective population size (*Ne*) is the number of breeding individuals in a population subjected to random selection and mating. The *Ne* is one of the most important foundations for evaluating the degree of endangerment for a specific population since it serves as a measure for genetic diversity loss in previous generations [100]. As a result, severe genetic diversity loss in most dog breeds is disastrous for an effective population size [19], which is an idealized size of the dog population based on genetic diversity loss [157]. In the last 100 years, population bottlenecks caused by historical events, as well as the establishment of closed studbooks, have shaped the formation of modern dog breeds [158]. As a result, the *Ne* of many modern breeds was reduced to <100, whereas the *Ne* of <50 is at high danger of inbreeding's harmful effects [159,160]. Reduced *Ne* is mainly caused by population substructures (produced by mating strategies, breeding aims, or geographical distances) influencing the increase in *F* [154]. Furthermore, low *Ne* causes much genetic drift and makes it difficult for the population to respond to changing environments [161].

A pedigree analysis by Machová et al. [67] revealed the *Ne* of Czech Spotted dogs to be as low as 10.28, causing an increase in inbreeding levels (36.45), and putting its existence in danger. Navas et al. [65] also reported reduced genetic variability in the Ca Mè dog population due to a reduced *Ne* of 13.25. The *Ne*, which was assessed at 21.76 for Polish Greyhounds [81], and 28.51 for Polish Hunting Dogs [104], had a similar lower value. Furthermore, Wijnrocx et al. [23] discovered that nine of the 23 dog breeds studied in Belgium had *Ne* values below 50, while seven had values between 50 and 100. Small *Ne* is associated with heterozygosity loss or loss of alleles [162]. The loss in genetic diversity may have developed from improvement programs aimed at selecting desired traits, leading to higher levels of inbreeding and the loss of founder alleles, which can result in undesired traits and health issues [4,33,85,163]. Therefore, the effective strategy for increasing *Ne* is to have an equal ratio of males and females. Greater importance should be placed on increasing the number of females because this is the issue that affects the population's growth [162].

*5.3. The Levels of Inbreeding in Dog Populations*

The inbreeding coefficient (*F*) represents an animal's probability of being homozygous for a locus by descent [164]. Inbreeding and relationship analyses, as well as their variations over time, have frequently been used to trace the evolution of genetic diversity in populations [56]. The inbreeding value describes a shift in a population's genetic structure in favour of homozygosity of gene sets at the expense of heterozygosity of the gene pool of individuals, implying a loss of genetic variability that can negatively impact fitness characteristics and increase the incidence of phenotypic defects [165]. Due to closed registries and breeding techniques, it is often believed that pedigree dogs are highly inbred, which has harmed the health and welfare of many pedigree breeds [159]. Several concerns about the rise in inbreeding and its effects on productivity, reproduction, and health in most livestock species, such as cattle, horses, sheep, and pigs, have lately been raised [166]. According to Kania-Gierdziewicz and Gierdziewicz [167], recent inquiry results show that increasing inbreeding levels has a negative impact on fertility, health, and productivity of many animal populations, particularly in less popular breeds with small populations. Kania-Gierdziewicz et al. [103] also reported that there is a real risk of inbreeding depression in the population of Tatra Shepherd dogs, due to the reduced effective number of founders and effective number of ancestors. As a result, health and reproduction issues could arise because of this situation.

Concerns about inbreeding's effects on heterozygosity, depression, the spread of hereditary disease, performance, welfare, and reduced diversity in dog breeds have led to a need for better monitoring and controlling practices [1,21,81,91,100]. Inbreeding's negative effects, such as lower productive and reproductive values within herds, are due to inbred individuals' lack of stability in dealing with environmental changes, making them

more vulnerable and susceptible [52]. Increased inbreeding in a dog population may cause not only health or fertility issues but may also influence litter size and litter composition (number of male and female puppies) due to early stillbirths of particularly male embryos or foetuses [166]. A high occurrence of physical diseases, defects, and disorders is a growing problem in many dog breeds [168], resulting in physical problems that affect individual dogs at young ages [169], and veterinary treatments are frequently required throughout life [170]. Hip dysplasia was also found to be associated with inbreeding depression in Icelandic sheepdogs [171], and in Tatra Shepherd dogs because of the limited number of breed representatives [103]. Furthermore, when modelling the most likely trend in the transmission of genetic risk for the disease over generations, a population of Dutch Labrador retrievers (with elbow dysplasia) showed estimates of relatedness to seven related ancestors [32]. One potential source of these issues is the loss of genetic variation and inbreeding, both of which are known to reduce individual viability [31].

Inbreeding can be estimated using pedigree information, which was widely used to calculate the $F$ within individuals and breeds [87]. However, unless estimations include the complete pedigree, the exact inbreeding may be underestimated by 5–10 fold [22]; [147,158,172,173]. Without control of inbreeding, genetic improvement can lead to a loss of genetic variability, decreasing the possibility for genetic gains in domestic populations [174]. As a result, good management of farm animal resources requires a thorough understanding of breed characteristics, such as population size and structure, geographic distribution, and genetic variation within and between breeds [49]. Furthermore, national breed organizations are researching ways for measuring genomic diversity to advise on breeding decisions and reduce disease prevalence while preserving desirable breed qualities and genetic diversity [94].

According to FAO [175], acceptable inbreeding rate should be between 0.5–1% per generation. Animals with an $F$ of 6.25%, on the other hand, are inbred and thus unacceptable [176]. With random mating, inbreeding increases at a rate of $1/(2Ne)$ per generation in an effective population size. For instance, inbreeding increases by 5% per generation in an effective population size of 10 [177,178]. According to Pekkala et al. [179], inbreeding and genetic drift both rise in small populations because there are fewer breeding individuals contributing to each generation. The loss of genetic diversity in closed-pedigree dog breeds can be a sensitive subject, leading to calls for open-registry [93]. The rate of inbreeding per year in Bullmastiff dogs reported by Mortlock et al. [94] was 1.2%, and Doekes [180] also reported inbreeding exceeding 1% in the Markiesje and Stabyhoun populations, which fell out of the range recommended for the inbreeding rate per year, implying a loss in effective population size. On the contrary, a study by Velie et al. [93] reported a low level of $F$ of 0.049 in the Australian Working Kelpie population, with an increase $F$ of 0.0016 per year, implying that opening a breed registry has a beneficial impact on the level of inbreeding within a population over the long term, as well as increased public awareness of the problems related to inbreeding and genetic diversity.

## 6. Genetic Diversity Conservation Strategies

General guidelines for the conservation of endangered populations [80,92,155,181].

- A genetically distinct population with high levels of genetic diversity, and a high level of adaptation and survival. It is recommended that if $Ne$ is greater than 500–1000 and not declining, changing management practices may not be necessary. Efforts should be dedicated toward preventing the formation of genetic bottlenecks, and the management programme should be developed to identify the number of individuals who can be sustained in the long run, allowing the population to be managed without major changes in population size.
- A genetically distinct population with high adaptation and survival yet low genetic diversity. Although it has a high level of adaptability, it is recommended to introduce some gene flow to the population to enhance $Ne$ and genetic variation. A good management strategy should also reduce the differences in individual contributions to the upcoming generation. If parents have different numbers of offspring, those who

have more offspring contribute a greater share of the genome transferred. Individuals who do not have offspring do not pass on allelic variations. As a result, a reasonable approach is to equalise individual contributions.

- A population that is distinct yet poorly adapted, with high levels of genetic diversity. It is recommended to determine why this population is poorly adapted and begin to select for improved local adaptation if the population is controlled. If a population does not react to selection, it is recommended to introduce populations with gene flow that is adapted to similar conditions.

- A locally adapted population that is not genetically distinct and has minimal levels of genetic diversity. It is recommended that management should promote improving *Ne* and genetic diversity. Furthermore, it is recommended to introduce populations with gene flow adapted to similar environments.

- A population that is not distinctive, poorly adapted, and has limited genetic variation. It is recommended to avoid mating of related individuals and introduce populations with gene flow that is adapted to the same conditions. Crossbreeding with a closely related breed may be an option for reintroducing genetic variation.

## 7. Conclusions

Pedigree information has been used in various studies to examine the genetic diversity of dog breeds. Pedigree information has proven to be an effective method for monitoring selection, breeding, and inbreeding changes over time, as well as conserving the diversity of dog breed populations. However, incomplete pedigrees limit the estimation of inbreeding in populations, preventing effective management of genetic diversity. Low EqG values were found in the Tetra Shepherd dog (3.44) in Poland, the Bullmastiff dog (3.24) in Australia, and Bichon fries (2.14) in Belgium, indicating the need to improve the quality of pedigree information to accurately estimate inbreeding and relationship coefficients.

Because pedigree analysis has many limitations in comparison to molecular markers, genomic studies must also be conducted to avoid pedigree errors, underestimation, or overestimation of genetic diversity parameters, and to assess the length of time inbreeding has influenced genetic diversity within a breed to effectively implement the best conservation strategies. All the STR and SNP panels under review have shown usefulness in evaluating genetic diversity in dog breeds as indicated by high $H_O$ (0.816) in African painted dogs in the United States of America using 14-STR. Using 20K SNP chip, the Sapsaree dog breed in the Republic of Korea had a $H_O$ of 0.342, which was high. Because of their high polymorphism, these molecular markers can effectively estimate genetic diversity in dog breeds even in the absence of pedigree information, regardless of the STR or SNP panel used.

According to the literature review, many dog breeds are losing genetic diversity in comparison with the original founder population due to reduced population sizes (genetic bottlenecks) and unequal use of breeding animals, all of which contribute to inbreeding and genetic diversity loss in dog breed populations. Inbreeding depression produces genetic abnormalities and infertility in dog breeds, compromising population growth, adaptability, and survival. As a result, effective genetic conservation strategies are needed to mitigate the reduced effective population size. It is necessary to increase the number of breeding animals and avoid mating of related animals and the use of popular sires.

Because breeding populations of dogs are not in full control, breeders make their own decisions, and the selection and use of sires may be unbalanced, it is important to measure the genetic diversity parameters for each generation.

**Author Contributions:** Conceptualization, writing of original draft and editing, R.S.M.; supervision, conceptualization, editing and structure for content, B.M.; supervision, conceptualization, editing and structure for content, M.L.M.; supervision, and conceptualization, K.A.N. All the authors have contributed and worked together on the manuscript. All authors have read and agreed to the published version of the manuscript.

**Funding:** This research was funded by the Department of Agriculture, Land Reform and Rural Development (DALRRD), grant number P02000187.

**Institutional Review Board Statement:** Not applicable.

**Acknowledgments:** Thank you to the Tshwane University of Technology and Agricultural Research Council.

**Conflicts of Interest:** The authors declare no conflict of interest.

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
