# Peer review of "Evaluation of Genetic Diversity in Dog Breeds Using Pedigree and Molecular Analysis: A Review"

_diversity, doi:10.3390/d14121054_

Round 1

Reviewer 1 Report

This is a very valuable overview for scientists, students and dog breeders. I have a few important remarks that need attention of the authors. 

1. The main problem in dog breeding is inbreeding and as a consequence the incidence of genetic defects. A perfect illustration is the omia website. Now the motivation to pay attention to genetic diversity is on adaptation instead of on inbreeding that gives rise to genetic defects and depression of fitness traits.

2. Frequently, the term conservation is used while management of genetic diversity is meant:  selection and mating of parents based on relationships.

3. The term bottleneck should be defined and there are three bottlenecks in dog breeding instead of two: 1) from wolf to dog, 2) the foundation of the breeds is often based on a limited number of parents and 3) the use of popular sires after an intense selection for exterior traits.

4. Somewhere in the paper (at the end?) it should be mentioned that breeding populations of dogs are not in full control: breeders take their own decisions, and the selection and use of sires may be unbalanced.  Therefore, the genetic diversity parameters should be calculated for each generation. 

5. At the end I should give the conclusions of this review for the use of pedigrees, STR's and SNP's. The recommendations are far too long and can be summarized and limited to the advises to avoid the loss of genetic diversity valid for all types of populations.

Minor remarks:

Line 1: on should be in.

Line 60: and in inbreeding with the occurrence of genetic defects and depression in fitness traits

Line 71: And the owner should give appropriate care to the dog and realizing that it is a descendance of the wolf and should have the possibility to show its natural behavior. 

Line 124: and the combination of different alleles into haplotypes that characterizes different breeds.

Line 183: Not true, also cryo-conservation can be used in dog breeding.

Line 192 and 193 should get a place in the introduction.

Line 207: Why are molecular markers better?

Line 257: overestimation of what?

Line 276: molecular markers can also be used to detect recent inbreeding (long ROH's)

Line 473: the use of popular sires causes the loss of founder alleles

Line 547: the emphasis should be on the increase of inbreeding per generation. That determines the effective population size.

Line 598: level should be inbreeding rate.

Author Response

Dear Reviewer,

Kindly see the attached document in response to the comments.

Thank you.

Reviewer 2 Report

This is an interesting review of the literature on the genetic diversity of dog breeds, and will be useful for all readers interested in the subject.

The text is well written, and contains only a small number of problems:

line 238: superfluous fragment: "According to Velie et al. (2021)"

line 299: Michels & Distl, 2020)

line 604: reported a low level

line 627: It is

ine 634: It is

line 638: It is

line 669: the Sapsaree dog .... had a Ho

lines 693-1499: References. Contain a mix of full and abbreviated journal titles which are often inconsequently written in large and low cases. All journal titles should follow the same system.

Author Response

Dear Reviewer,

Kindly find the attached document in response to the comments.

Thank you.

Reviewer 3 Report

Dear authors

This is a very interesting topic and you have done a very well job in researching the literature for information. I have the following comments for the manuscript:

Line 42: can you mention some selected behaviors?

Line 55-56: Between and within breeds is enough, remove the sentence "between individuals ..."

Lines 81-83: Some features (e.g. narcotics, bombs, are repeated with the previous section), please rephrase

Line 128: as well as the proportions in which these changes occur. Please explain

Lines 330-371: In my opinion, the programming languages that were used for the development of application or the run time and computer characteristics have nothing to add to the paper, since it is not a technical paper. Please remove these info and provide only the important features (p.ex. links to their sites)

Line 406: according to a study (not Italian, maybe the contributors were of different origin)

Lines 495-497: bottlenecks are mentioned in previous sections, it has to be removed

How will affect a gene flow the selected phenotypes of the animals? It is not analysed in the MS. Are there any examples of introduction of individuals? and how this introduction affected the indices of the populations?

Author Response

(The authors gave the same response as above.)
